# Telovelar Approach for Fourth-Ventricular Epidermoid Cyst: Anatomical Respect, Functional Recovery, and Long-Term Stability

**DOI:** 10.3390/diagnostics16010129

**Published:** 2026-01-01

**Authors:** Cosmin Pantu, Alexandru Breazu, Stefan Oprea, Mugurel Petrinel Rădoi, Octavian Munteanu, Nicolaie Dobrin, Catalina-Ioana Tataru, Alexandru Vladimir Ciurea, Adrian Vasile Dumitru

**Affiliations:** 1Department of Anatomy, “Carol Davila” University of Medicine and Pharmacy, 050474 Bucharest, Romania; cosmin.pantu@umfcd.ro (C.P.); octavianmunteanu@umfro.com (O.M.); 2Department of Medical Research, Puls Med Association, 051885 Bucharest, Romaniaprof.ciureaav@umfcd.ro (A.V.C.); 3Department of Neurosurgery, “Carol Davila” University of Medicine and Pharmacy, 050474 Bucharest, Romania; 4“Nicolae Oblu” Clinical Hospital, 700309 Iasi, Romania; 5Clinical Department of Ophthalmology, “Carol Davila” University of Medicine and Pharmacy, 020021 Bucharest, Romania; 6Department of Ophthalmology, Clinical Hospital for Ophthalmological Emergencies, 010464 Bucharest, Romania; 7Neurosurgery Department, Sanador Clinical Hospital, 010991 Bucharest, Romania; 8Medical Section, Romanian Academy, 010071 Bucharest, Romania; 9Department of Pathology, Faculty of Medicine, “Carol Davila” University of Medicine and Pharmacy, 030167 Bucharest, Romania

**Keywords:** fourth ventricle, epidermoid cyst, telovelar approach, microsurgical anatomy, posterior fossa, cerebellar ataxia, cerebrospinal fluid dynamics, neurophysiologic monitoring, aseptic meningitis, functional recovery

## Abstract

**Background and Clinical Significance**: Fourth-ventricular epidermoid cysts are rare intracranial lesions. They account for fewer than 1% of all primary brain tumors. Fourth-ventricular epidermoid cysts grow slowly because they are closely related to brainstem, cerebellum, and major blood vessels, so their treatment requires special caution. Because the cyst capsule attaches to functionally sensitive locations, complete removal is usually not possible without compromising some aspect of brain or spinal cord function. Surgical decision-making always involves weighing the need to remove the entire cyst against the potential loss of function of the affected area. The following case study describes how a patient was treated with a focus on the relationship between the cyst and surrounding anatomy, allowing for successful decompression with minimal risk to the patient’s neurologic status. **Case Presentation**: A young adult female patient was hospitalized with progressive truncal ataxia, disequilibrium and occipital headache accompanied by papilledema. Her physical examination disclosed significant dysfunction of the midline cerebellar region (SARA score = 18/40, ICARS score = 42/100), gaze-evoked nystagmus and bilaterally elevated grade II papilledema. MRI and MRA demonstrated a large, lobulated, nonenhancing, avascular mass located within the fourth ventricle, encroaching upon the dorsal medulla and obstructing both the foramen of Magendie and foramina of Luschka—findings typical of an epidermoid cyst. Microsurgical resection was accomplished via a median suboccipital craniectomy using a telovelar approach along the embryonic cerebellomedullary fissure to protect the integrity of the vermis and brainstem. The cyst contained layers of keratin embedded in a thin, translucent capsule. The capsule was carefully dissected away from the floor of the fourth ventricle. A very narrow band of capsule attached to the rhomboid fossa was intentionally spared to avoid damaging the cranial nerves. The patient had normal cerebrospinal fluid circulation restored and normal ventricular pulsation observed during surgery. Histopathology confirmed a benign epidermoid cyst consisting of keratinizing stratified squamous epithelium containing cholesterol clefts and laminated keratin debris. After surgery, the patient exhibited continuous neurological improvement including restoration of balance, disappearance of her headaches, and normalization of ocular pursuit. Sequential imaging studies were conducted post-operatively at one week, one month, three months, five months, and seven months to document stable decompression of the fourth ventricle, re-expansion of the fourth ventricle, and no evidence of cyst recurrence. Post-operative course was uncomplicated and the patient has remained free of symptoms and fully independent functionally at most recent follow-up. **Conclusions**: This case illustrates that when anatomically oriented, “maximal safe resection” can result in long-lasting decompression and clinically meaningful improvement in neurological function in patients with fourth-ventricular epidermoid cysts. Restoration of the patient’s natural cerebrospinal fluid pathway and preservation of neural interface relationships is more beneficial than pursuing aggressive removal of the cyst capsule. Although the risk of late recurrence is present even after nearly total removal, continuous radiologic monitoring is necessary to identify any recurrence. These experiences illustrate that with the principles of surgical restraint and anatomical guidance, there can be a balance between long-term stability and low operative risk.

## 1. Introduction

Cysts of the posterior fossa located centrally in the midline are very rarely seen as benign congenital inclusion cysts formed by ectodermal remnants that fail to degenerate after the closing of the neural tube. As these cysts grow gradually and along the cerebrospinal fluid (CSF) spaces and cisterns of the posterior fossa, and thus tend to deform to the adjacent structures without invading them, the patient generally will not experience any symptoms until there is sufficient mass effect to either compromise the circulation of CSF or to disrupt brainstem-cerebellar circuitry [1]. Most posterior fossa epidermoids are found in the cerebellopontine angle (CPA) and while many are found in this area in a lateral position, pure midline posterior fossa epidermoids are extremely rare, and those found in association with the fourth ventricle are a very small percentage of all reported cases [2].

The clinical characteristics of central midline posterior fossa epidermoids will primarily be related to the mass’s relationship to the fourth ventricle and its exit tracts. Indolently growing cysts can lead to obstruction of CSF flow resulting in hydrocephalus and papilledema, and also can produce a fastigial/vermian syndrome selectively based upon distortion of the roof of the ventricle and/or compression of the cerebellomedullary fissure [3]. Fourth-ventricle masses in adults can be ependymomas/subependymomas of the ventricular floor, rosette-forming glioneuronal tumors of the superior vermis, exophytic dorsal brainstem gliomas, and various other cystic lesions that can mimic CSF signal. Diffusion weighted imaging (DWI) is the best sequence for identifying epidermoid cysts and for monitoring them due to the high degree of diffusion restriction that it demonstrates that remains on follow up and can distinguish epidermoid cysts from arachnoid cysts and virtually all other neoplastic cystic lesions [4].

Recent studies have proposed an epicenter-based posterior fossa midline epidermoid tumor (PFMET) classification system to better predict the anatomy of the surgical field and to assess the risk associated with surgery for midline posterior fossa epidermoid cysts. The classification system divides PFMETs into three categories: Type 1—with an epicenter in the cisterna magna; Type 2—with an epicenter in the fourth ventricle; and Type 3—with dual epicenters spanning the cisterna magna and fourth ventricle [5]. The classification system provides additional information regarding the expansion patterns of midline epidermoids along embryonic planes that involve the inferior medullary velum and tela choroidea and that allow the cyst to enter or traverse the fourth ventricle through the cerebellomedullary fissure. Identifying these relationships on MRI is useful as Type 2 and Type 3 lesions, by virtue of their direct roof-to-floor ventricular contact, have a greater tendency to adhere focally to the rhomboid fossa and lateral recess/outlet region, which represent the two areas at greatest risk for cranial nerve or bulbar complications during dissection [6].

While gross total resection is considered the goal of therapy for posterior fossa epidermoid cysts, studies have demonstrated that adhesion burden and post-operative risk can often be predicted by pre-operative factors. Studies have shown that larger tumor volumes, longer durations of symptoms prior to surgery, and older patients are independent predictors of capsular adhesion burden and post-operative risk of cranial nerve deficits, chemical meningitis, and transient bulbar dysfunction [7]. These predictive factors are currently evaluated using the PFMET framework, as the epicenter and growth corridors predict the site of adhesion surfaces. Therefore, current evidence supports a risk-adapted approach to surgery that avoids aggressive pursuit of the capsule when the dissection plane approaches the rhomboid fossa [8].

With these concepts in mind, we present a case report of a lesion most consistent with a Type 2 PFMET (fourth-ventricle epicenter) treated via a fissure-based telovelar approach. This case illustrates (i) how preoperative MRI evaluation of the posterior fossa, specifically the use of DWI to evaluate diffusion restriction, can provide prospective identification of midline subtype and delineation of the relationships between the tela choroidea and the cerebellomedullary fissure; (ii) how these planes are translated intra-operatively to a roof-dominant interface with a narrow high-risk adhesion band at the ventricular floor; and (iii) how limiting the resection at this adhesion zone can result in long-term neurological recovery and diffusion-negative follow-up after nearly complete resection.

## 2. Case Presentation

The patient was a 28 year old woman who had no prior medical history. The patient came into the hospital with progressive imbalance and difficulty with her gait over a few months. The symptoms started with mild disequilibrium and occasional swaying, but eventually evolved into constant unsteadiness and over the last month she was unable to stand alone. At the same time, the patient experienced morning predominant occipital headaches that gradually went from mild to severe. They were worsened by the Valsalva maneuver and/or coughing, or when bending forward with her head. Occasionally, these episodes would be followed by projectile vomiting that lasted momentarily and provided transient pain relief. In addition, during the past week the patient complained of posterior cervical stiffness and photophobia without fever, trauma or systemic illness. Because of the gradual progression of symptoms with episodic intracranial pressure (ICP) elevations, the clinical presentation was suggestive of an indolently growing posterior fossa mass as opposed to inflammatory, vascular or metabolic diseases.

When she was admitted, she was alert and awake and her Glasgow Coma Score was 15/15. Due to marked instability, she moved and acted in a manner that was guarded and intentional. A midline cerebellar syndrome predominated her neurological examination. When asked to rise from bed without assistance, she was able to remain upright for approximately three to four seconds before she fell backward. There was continuous truncal oscillation (approximately 3–4 Hz, 2–3 cm amplitude) which indicated vermian dysfunction. Romberg testing was abnormal. Her gait was broad-based, unsteady, and unpredictable and she was unable to walk in tandem. Her SARA score was 18/40, which indicates moderate to severe axial ataxia, and her ICARS score was 42/100, which confirms that her axial symptoms are predominant with minimal involvement of her limbs. Coordination of her upper limbs was almost completely preserved and she demonstrated mild terminal dysmetria with finger-to-nose and heel-to-shin testing and there was no evidence of tremors or rebound. Muscle strength and tone were normal (MRC 5/5), reflexes were brisk and symmetrical and she had no sensory abnormalities. Although she would sway while seated, she ceased doing so immediately upon lying down, which supported the presence of central instability of the vermian lobe. Mildly scanning speech was noted. Full eye movements were present; however, smooth pursuit was interrupted and horizontal gaze-evoked nystagmus occurred after five to six beats and hypometria of saccades with unstable fixation were noted which suggested involvement of the flocculonodular pathways. All cranial nerves IX–XII were intact and there was no indication of any swallowing or articulatory abnormalities. Her NIH stroke scale was 2 (limb ataxia), and her modified Rankin Scale was 3. Her Barthel Index was 70/100 and her Karnofsky performance status was 70, which reflect preserved overall health despite significant neurologic impairment.

There was clinical evidence of increased intracranial pressure. When she was passively flexed at the neck, she complained of immediate discomfort and occipital pain. Her headaches worsened with Valsalva maneuvers. She described transient visual obscuration while standing. Bilateral papilledema of Grade II on the Frisen scale with blurred margins of the optic discs, mild elevation, and venous engorgement without hemorrhage were observed on fundoscopic examination. Absence of venous pulsation was also noted. Her cardiovascular measurements were unchanged without bradycardia or hypertension. Her laboratory values were within normal limits and excluded infectious, metabolic, or inflammatory etiologies. The combination of truncal ataxia with relatively preserved limb function, papilledema, ICP-related headache with vomiting, and cervical stiffness identified a midline cerebellar-fourth ventricle process causing obstructive hydrocephalus.

A careful differential diagnosis was performed. Fourth ventricular ependymomas or subependymomas were considered based on their potential locations; however, these tumors usually occur earlier in life and have cranial nerve deficits and more rapid progression than what this patient experienced. Vermian astrocytomas are less common in younger patients and tend to have more hemispheric involvement. Arachnoid cysts can produce chronic hydrocephalus; however, the degree of axial ataxia and papilledema present in this patient suggested direct parenchymal compression. Based on her age, the long, slowly progressive course, the episodes of ICP crises, the strictly midline cerebellar syndrome, and the preservation of cranial nerves, a benign fourth ventricle epidermoid cyst was thought to be the most likely diagnosis and prompted an immediate imaging study.

The preoperative MRI demonstrated a multilobulated mass within the fourth ventricle which was molded to the roof and cavity of the fourth ventricle. The mass arose from the inferior portion of the vermis and draped over the superior medullary velum and had smooth interfaces with the surrounding parenchyma which it displaced but did not invade. The mass was uniformly hypointense on T1 images; however, on T2/FLAIR images, the mass had hyperintensity with internal lobulation. The mass marginally compressed the uvula and nodulus and compressed the posterior surface of the pontine-medullary tegmentum. The foramen of Magendie was obliterated and both foramina of Luschka were narrowed resulting in an upstream flow void in the ventricles and a rim of periventricular interstitial edema. This created the hydrodynamic environment responsible for the patient’s symptoms. The worsening headache and vomiting in the morning, the papilledema, and nuchal rigidity resulted from the hydrodynamic environment. The avascular nature of the mass and the CSF conforming surface of the mass, as well as its central location, strongly suggested the possibility of an epidermoid cyst vs. a vascular neoplasm. The image-symptom correlation was excellent: the axial ataxia and titubation were due to compression of the fastigial nucleus and the vermian midline; the gaze-evoked nystagmus was due to disruption of the flocculonodular circuitry along the fourth ventricular roof; and the intracranial hypertension was due to obstruction of the caudal apertures. The morphology and epicenter of the mass were most consistent with a posterior fossa midline epidermoid tumor of Type 2 (fourth-ventricle epicenter) with a dominant relationship to the tela choroidea and cerebellomedullary fissure, anticipating a roof-centered growth pattern and possible focal adherence to the ventricular floor.

On the axial post contrast T1 plane (Figure 1A), the mass occupied the ventricular space like a mold, did not enhance, and was in the midline. The medial surface of the mass was in contact with the cerebellar vermis and the lateral surface of the mass slightly indented the tonsillar surfaces without penetrating into the parenchyma. On the mid sagittal T1 view (Figure 1B), the mass was in contact with the inferior vermis and caused an upward bowing of the superior medullary velum and flattening of the dorsal medulla and this explained the patient’s episodes of pressure-induced vomiting by mechanical stimulation of the area postrema. The coronal T1 reconstruction (Figure 1C) clearly demonstrated that the mass was in the midline and caused an equal compression of the two outlet recesses and was consistent with the absence of laterality in the patient’s deficits. On T2 (Figure 1D) and mid sagittal FLAIR (Figure 1E), the mass’ fluid-like appearance and internal lobulation were prominent; the surface of the mass was molded to the fourth ventricular CSF and not invading the fourth ventricular CSF, and this was a classic finding for this type of mass and accounted for the unusual severity of the fastigial/vermian hub being compressed and not destroyed.

In addition to its rarity, the case also stood out due to the fact that all of the patient’s preoperative angiograms revealed a normal architecture of cerebral vasculature and no indication of aneurisms, arteriovenous malformations, or impaired venous outflow (Figure 2A,B). Further, the Circle of Willis was fully developed and symmetrically distributed, and the dural venous sinuses were unobstructed. As such, these studies ruled out vascular-related causes of intracranial hypertension and were consistent with a slowly expanding, avascular lesion.

Radiologically, the tumor was identified as a nonenhancing, avascular midline fourth-ventricular mass compressing the dorsal brainstem and resulting in secondary supratentorial ventriculomegaly. The intraoperative anatomy was correctly predicted: the lesion was avascular, white, waxy and was 39 × 37 × 55 mm in size.

To immobilize the head and provide a linear pathway to the fourth ventricle, while maintaining venous drainage, a general anesthetic regimen was employed utilizing a three-pin skull clamp to flex the occipitocervical joint. The bony suboccipital squama was exposed via a midline incision and a median suboccipital craniectomy was performed along the midline working axis. The dura mater was tight and was therefore widely opened in a star-shaped manner to immediately achieve CSF decompression and relaxation. Under microscopic examination, the inferior vermis was found to be thinned and elevated. The cerebellomedullary cleft and tela choroidea were opened to gain access to the fourth ventricle via a telovelar route; no vermian corticectomy was performed except for that necessary to obtain access via the fissure plane. Beneath a very delicate layer of arachnoid, a pale opalescent surface denoted the location of the lesion. The arachnoid was then opened sharply without coagulation.

The cyst wall was pearly, avascular, and characteristic of an epidermoid. The laminated keratin was removed in small pieces using low pressure suction and fine ring curettes, with multiple irrigations to prevent loose flakes from developing into chemical meningitis. Decompression restored the ventricular space, allowing visualization of the ependymal lining and vermian leaflets. The dissection continued around the periphery of the cyst wall, only removing it when a clear plane existed. Gradually, the floor of the fourth ventricle and its associated landmarks became visible without the use of traction. The most significant interface encountered was between the tela choroidea/inferior medullary velum complex, which was consistent with a Type 2 midline lesion. However, a thin segment of the capsule remained firmly attached to the rhomboid fossa; since no safe cleavage plane could be identified, the decision was made to leave the remaining portion of the capsule in place to avoid injury to the brain stem. Post-decompression, the fourth ventricle re-expanded completely, and Magendie’s opening was visualized to have re-opened directly, and both Luschka openings were patent with free CSF flow. Minimal hemostasis was achieved with only two brief bipolar applications and application of Surgicel. The dural defect was left unsutured to avoid re-tensioning, and the soft tissue layers were closed anatomically.

Upon completion of surgery, the patient was extubated in the OR and transported to ICU for routine post-operative monitoring. Rapidly, the patient regained full consciousness and orientation (GCS = 15) and began speaking spontaneously without dysarthria. Serial neurologic examinations conducted throughout the first evening confirmed that the patient had intact ocular alignment, cranial nerves and normal limb motor function. Transient positional disequilibrium decreased during the initial days. Headaches resolved and nausea and vomiting did not recur indicating normalization of ICP. Neck motion improved by 24 h and the patient was able to sit upright without oscillation. By the third post-operative day, the patient’s gait had narrowed and became consistent, and the patient walked with minimal assistance. At the time of discharge (Post-operative day 8), mRS improved to 1 and SARA to 4/40.

CT scans completed on the first post-operative day (Figure 3) were utilized to exclude hemorrhage and assess early ventricular diameter; they indicated posterior fossa re-expansion, full reopening of the fourth ventricle, normal brainstem convexity, and no early hydrocephalus. A second CT scan completed 7 days post-operatively (Figure 4) verified stable ventricular and outlet patency and excluded late onset hydrocephalus. The one month follow-up CT scan (Figure 5) provided a baseline for long-term comparisons. Later surveillance relied upon MRI including DWI to exclude diffusion-positive residual or recurrence.

The histopathological examination confirmed that the tumor was an epidermal cyst with an intra-cerebral location; the cyst contained layers of keratinous debris, separated by stratified squamous epithelial cells which lacked atypical characteristics or inflammation. At her one month post-operative follow-up visit, the patient’s clinical condition was essentially normal. She could walk without deviation from center line, had eliminated trunk sway, had normal speech, had regained normal pursuit movements and had no gaze evoked nystagmus. She denied having headaches, vision changes, or neck stiffness.

Follow up CT scans three months after surgery (Figure 6), confirmed continued decompression of the fourth ventricle and structural stability, as evidenced by stable size of the fourth ventricle, open CSF pathways, and absence of inflammation or scarring. The patient was clinically normal and able to resume all normal activities

MRI scans five months after surgery (Figure 7), demonstrated that the post-operative cavity contained fluid with no enhancement and normal perfusion of the surrounding parenchyma, with no glial scar on FLAIR and no diffusion restriction on DWI, effectively ruling out remaining keratinous material.

CT scans seven months after surgery (Figure 8), demonstrated long term stability: the fourth ventricle remained widely patent, the posterior fossa proportions were normal, the CSF spaces were symmetric, and there was no evidence of recurrence, new calcification, subdural collection, or ventricular change. Clinically she remained neurologically intact, describing stable gait and effortless balance. Final scores were MRS = 0, SARA = 1/40, and she returned to work and daily activity.

Throughout follow-up at 1 week and at 1, 3, 5, and 7 months, the patient demonstrated continuing clinical recovery and stable radiographic restitution with no diffusion positive residual or recurrence. The sustained normalization of CSF pathways and posterior fossa anatomy represents the effectiveness of a fissure-based telovelar surgical approach using embryonic planes and judicious preservation of a focal adherent capsule remnant at the rhomboid fossa to preserve neural safety.

## 3. Discussion

Fourth-ventricle midline epidermoids pose a particular challenge to surgeons due to the limited space available to dissect around the ventricular floor. In addition, since outcome from surgery is dependent upon whether the surgeon has sufficient information to accurately identify the plane beyond which dissection poses undue risk, our case exemplifies the utility of the PFMET framework to provide such information and support a surgical approach that balances maximal resection with minimal risk of complication [9].

Epicenter-based PFMET maps are only useful to surgeons if they change the way a surgeon thinks about the anatomy of the area involved in surgery. For example, prior to surgery, MRI indicated that this patient’s fourth-ventricle midline epidermoid had grown predominately on its roof via the tela choroidea inferior medullary velum corridor, extending into the cerebellomedullary fissure [10]. As a result of this type of growth pattern, we expected to find significant adhesions at the rhomboid fossa but not at the many broad cisternal neurovascular interfaces; this expectation was confirmed during surgery. The roof was found to be primarily adherent and separable, whereas there were only very localized dense adhesions to the ventricular floor. This concurs with the reviewer’s comment that the PFMET map gives a better appreciation of the relationship of the anatomy to the areas of potential difficulty in surgery that can be appreciated on MRI [11].

Prior studies have shown that the presence of capsule adhesions and complications following surgery for posterior fossa epidermoids are correlated with preoperative variables including larger tumor sizes, longer durations of symptoms and older ages of the patient. Additionally, in the original PFMET study of multiple cases of posterior fossa epidermoids, those cases that had brainstem adhesion demonstrated a significantly greater size and duration of symptoms compared to cases without brainstem adhesion and Type 3 tumors demonstrated the highest rate of adhesion and postoperative risk [12].

This patient was younger than most cases studied in prior investigations, however the size of the tumor and duration of symptoms were typical for a case of posterior fossa epidermoid, therefore we believed that there was a reasonable basis for concern regarding the potential for adhesions at surgery. However, what PFMET provided for us was topographic prediction: although the preoperative data suggested that there would be an increased likelihood of adhesions, the data also suggested that the adhesions would be located at the rhomboid fossa; this distinction is important since an attempt to remove a significant portion of the capsule at the ventricular floor is the most common cause of postoperative morbidity related to lower cranial nerves or the bulbar region in patients undergoing surgery for posterior fossa epidermoids.

Residual capsule is a factor that contributes to recurrence of epidermoids, however the permanent morbidity that occurs with aggressive dissection of adherent surfaces of the brainstem is much stronger correlation with aggressive dissection. Since the rhomboid fossa is a fixed functional border in midline fourth-ventricle epidermoids, the decision to leave a thin layer of adherent capsule behind represents a choice based on the risk of aggressive dissection, rather than an incomplete operation [13]. Our patient’s long-term diffusion-negative MRI and complete functional recovery demonstrate that in Type 2 lesions, a restrained approach to resection at the predicted adhesion surface can achieve a balance between the risk of recurrence and the risk of neurological morbidity. The goal of Table 1 is to evaluate the significant clinical and anatomical findings that have influenced contemporary approaches to the treatment of fourth ventricle epidermoid cysts.

In midline fourth-ventricle epidermoids, fissure-based telovelar access utilizes the same embryologic pathways that define the growth patterns of the tumor. In our case, accessing the tumor via the cerebellomedullary fissure and tela choroidea allowed for a direct roof-first access, thereby providing decompression of the tumor prior to addressing the floor [24]. It is likely that this sequence of events minimized traction on the rhomboid fossa and facilitated the recognition of the safe vs. unsafe planes in the tumor capsule. The fact that the operative findings demonstrated a predominantly roof-dominant capsule interface further supports the use of telovelar corridors as the preferred method for Type 2 lesions, while reserving transvermian splitting for superiorly located epicenters or for corridors that cannot be safely accessed [25].

While the evidence base consists of small series, there appears to be a uniformity in the literature concerning the behavior of Type 2 midline fourth-ventricle epidermoids, including: (i) large intraventricular lesions typically present with truncal ataxia and/or obstructive symptoms of the outlet; (ii) roof-tela choroidea attachment is frequently the separable component of the tumor; (iii) floor adhesions tend to be focal, rather than widespread, and (iv) the best outcomes are achieved when near total resection of the tumor capsule preserves the floor plane when the adhesions are inseparable [3]. Our case follows this pattern and adds a complete radiologic-operative-clinical pathway that supports the predictive value of epicenter mapping.

Due to the rarity of fourth-ventricle midline epidermoids, all conclusions are derived from small cohorts with varying levels of follow-up. Nevertheless, within this limitation, this case supports a practical principle: preoperative epicenter mapping can predict the geography of adhesions and selective restraint at the predicted adhesion surface of the floor can lead to durable neurological recovery without visible diffusion residue. Should subsequent multicenter studies using PFMET-stratified datasets confirm that Type 2 lesions behave similarly, epicenter-based planning could become a standard tool for determining the appropriate surgical corridor as well as the maximum extent of capsule dissection.

## 4. Conclusions

Midline fourth-ventricle epidermoid cysts are rare lesions located in a surgically limited compartment whose operative goal is to restore fourth-ventricle/CSF outlet patency while simultaneously preserving the rhomboid fossa, its functional substrate, and all related lower cranial nerves. Our patient underwent extensive pre-operative MRI evaluation, which demonstrated a posterior fossa midline epidermoid cyst centered around the fourth ventricle (PFMET Type 2), and with a roof-dominant growth pathway extending along the inferior medullary velum–tela choroidea complex; based upon these findings, we hypothesized that adhesions potentially compromising CSF flow through the fourth ventricle would be focal at the level of the ventricular floor. These anatomical predictions were subsequently validated intra-operatively during tumor resection, wherein the dominant separable interfaces were those between the ventricular roof and the lateral walls of the fourth ventricle; however, a small portion of the tumor’s capsule could not be dissected free from the rhomboid fossa.

The telovelar fissure corridor provided an unobstructed route to access the tumor via native anatomic planes, thereby enabling a roof-first approach to safely decompress the tumor, allowing circumferential mobilization of the tumor capsule where it remained attached to clearly identifiable cleavage planes, and providing precise identification of the location of a solitary high-risk area of adhesion. Consequently, near-total resection of the tumor was accomplished without causing injury to either the bulbar musculature or the lower cranial nerves, due to deliberate preservation of a small portion of the tumor capsule at the level of the ventricular floor. Following surgery, the patient rapidly regained normal cerebrospinal fluid dynamics, and subsequent resolution of his ataxic symptoms occurred in conjunction with full functional recovery, including continued expansion of the fourth ventricle and absence of diffusion positive tumor residue on postoperative MRI studies.

Collectively, this case highlights the utility of epicenter-based midline classification systems as preoperative tools for predicting both the potential surgical planes and the geography of adhesions in fourth ventricle epidermoid tumors, and illustrates a risk-adaptive telovelar surgical strategy in which aggressive tumor resection occurs only within the boundaries of safe embryonic developmentally derived pathways and is halted at the site of any unsafe floor adhesions. Within the confines of a single clinical case report, the presence of stable radiographic and neurologic outcomes supports the premise that functional preservation of the rhomboid fossa and the lower cranial nerves is compatible with long-term successful management of epidermoid tumors when decisions regarding surgical intervention are made using anatomical prediction from preoperative imaging studies and adherence to anatomic planes during intraoperative procedures.

## Figures and Tables

**Figure 1 diagnostics-16-00129-f001:**
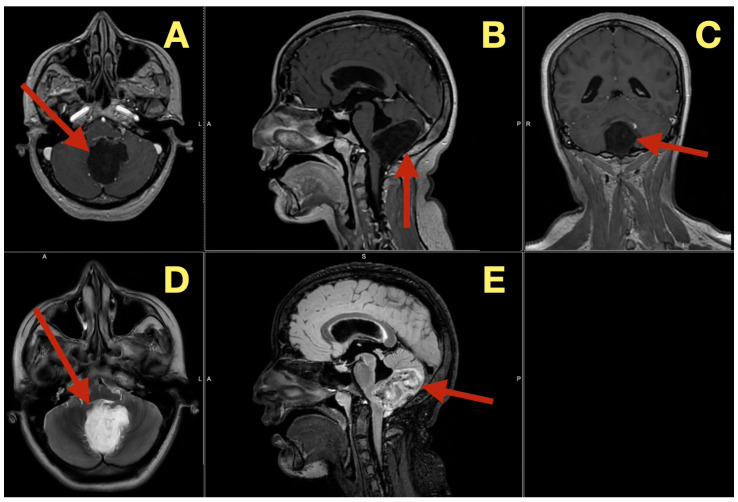
Preoperative MRI of the posterior fossa. (**A**): Axial post-contrast T1: non-enhancing, lobulated mass centered in the fourth ventricle (arrow). (**B**): Mid-sagittal T1: inferior-vermis contact with upward bowing of the superior medullary velum and dorsal brainstem compression (arrow). (**C**): Coronal T1: strict midline localization with symmetric crowding of the ventricular outlets (arrow). (**D**): Axial T2: intrinsically high signal with molded margins against cerebellar tonsils (arrow). (**E**): Mid-sagittal FLAIR: high signal lesion with effacement of Magendie and narrowing of Luschka, concordant with obstructive hydrocephalus (arrow). Radioclinical correlation: vermian compression → truncal ataxia/titubation; flocculonodular disturbance → gaze-evoked nystagmus; outlet occlusion → intracranial hypertension.

**Figure 2 diagnostics-16-00129-f002:**
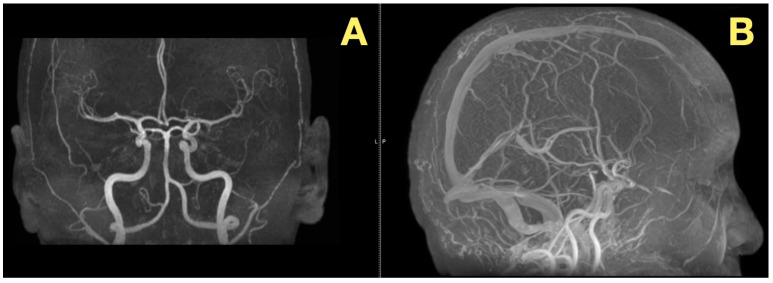
Preoperative vascular imaging. (**A**): Frontal MRA maximum-intensity projection: normal configuration of the circle of Willis without aneurysm or stenosis (**B**): Lateral MRA: preserved arterial flow and patent dural venous sinuses, excluding vascular or venous causes of intracranial hypertension; the pressure increase is attributable to CSF-outflow obstruction by the fourth-ventricular mass.

**Figure 3 diagnostics-16-00129-f003:**
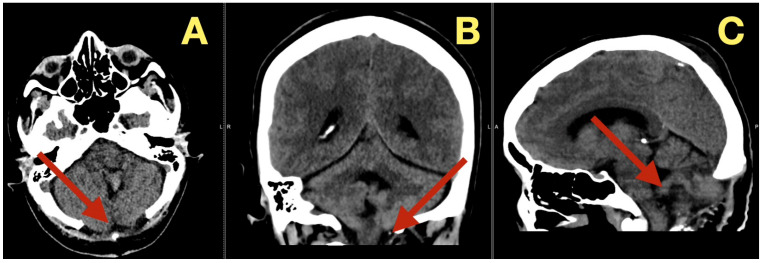
Immediate postoperative CT scan. (**A**): Axial CT showing full decompression of the fourth ventricle and restoration of normal CSF circulation (arrow). (**B**): Coronal CT demonstrating complete midline re-expansion with absence of residual mass (arrow). (**C**): Sagittal CT confirming brainstem relaxation and normalization of posterior-fossa anatomy (arrow).

**Figure 4 diagnostics-16-00129-f004:**
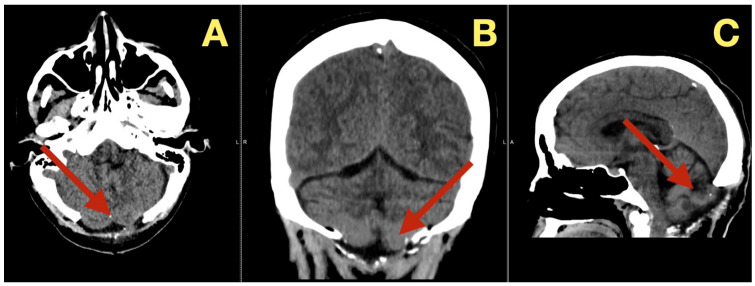
One-week postoperative CT scan. (**A**): Axial CT demonstrating a clean surgical cavity with complete re-expansion of the fourth ventricle and normal CSF flow through the foramen of Magendie (arrow). (**B**): Coronal CT showing preserved midline symmetry and restored perimedullary cisterns without evidence of fluid collection or residual density (arrow). (**C**): Sagittal CT confirming normalization of posterior-fossa configuration, intact vermian curvature, and patent foramina of Luschka (arrow).

**Figure 5 diagnostics-16-00129-f005:**
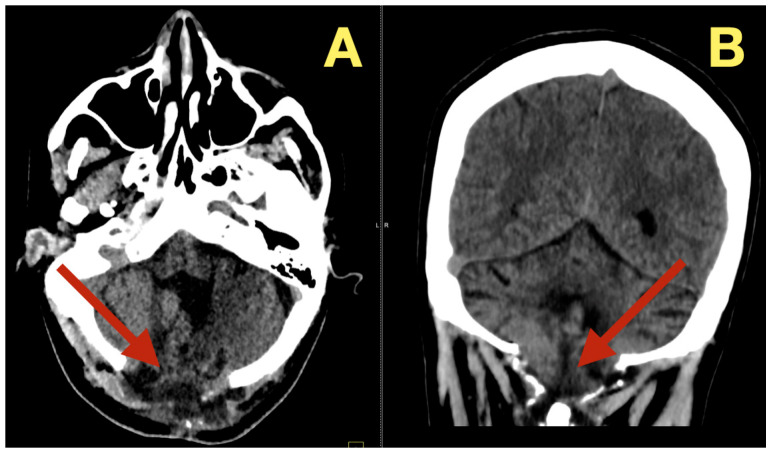
One-month postoperative CT scan. (**A**): Axial CT showing a well-aerated posterior fossa with a completely re-expanded fourth ventricle and stable CSF spaces, confirming durable decompression and absence of recurrence (arrow). (**B**): Coronal CT demonstrating preserved symmetry of the cerebellar hemispheres, intact vermian architecture, and normal perimedullary cisterns, indicating full restoration of posterior-fossa morphology (arrow).

**Figure 6 diagnostics-16-00129-f006:**
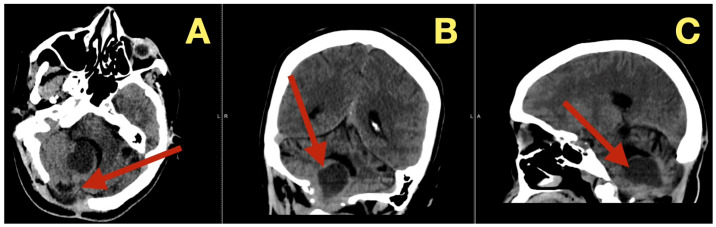
Three-month postoperative CT scan. (**A**): Axial CT revealing sustained decompression of the fourth ventricle and preservation of the posterior-fossa configuration (arrow). (**B**): Coronal CT showing symmetrical cerebellar folia and patent lateral recesses without evidence of recurrent mass (arrow). (**C**): Sagittal CT confirming persistent brainstem relaxation, patent foramen of Magendie, and absence of postoperative scarring (arrow).

**Figure 7 diagnostics-16-00129-f007:**
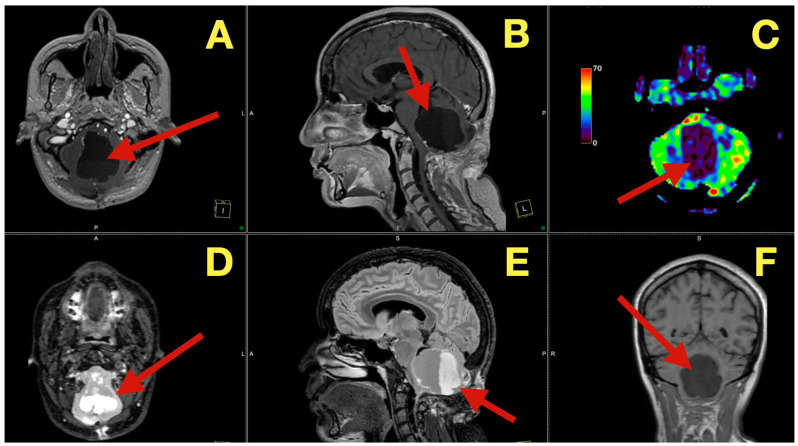
Five-month postoperative MRI. (**A**): Axial T1-weighted image demonstrating a CSF-filled postoperative cavity with smooth margins and no enhancing tissue (arrow). (**B**): Sagittal T1 image showing sustained patency of the fourth ventricle and normal vermian convexity (arrow). (**C**): Axial perfusion map revealing normal parenchymal hemodynamics in the cerebellum surrounding the resection site (arrow). (**D**): Axial diffusion-weighted image showing absence of restricted diffusion, excluding residual keratinaceous material (arrow). (**E**): Sagittal FLAIR sequence illustrating clean CSF spaces and absence of gliosis (arrow). (**F**): Coronal T1 image confirming midline symmetry and complete restitution of cerebellar and brainstem anatomy (arrow).

**Figure 8 diagnostics-16-00129-f008:**
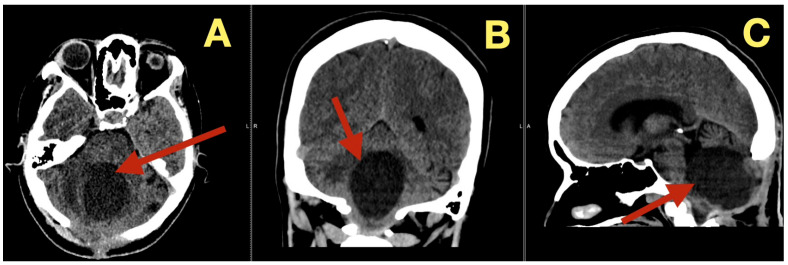
Seven-month postoperative CT scan. (**A**): Axial CT showing persistent decompression of the fourth ventricle and normal CSF distribution (arrow). (**B**): Coronal CT demonstrating stable midline anatomy and unchanged posterior-fossa dimensions with no residual lesion (arrow). (**C**): Sagittal CT confirming complete restoration of brainstem and cerebellar geometry, maintained CSF flow, and absence of new pathology (arrow).

**Table 1 diagnostics-16-00129-t001:** This table intends to summarize the pivotal clinical, anatomical, and technical studies that have shaped current strategies for the management of fourth ventricle epidermoid cysts. It highlights how progressive refinement—from trans-vermian to telovelar and endoscopic-assisted routes—has improved safety, reduced recurrence, and preserved cerebellar and brainstem function. Collectively, these studies reinforce the guiding principle of modern posterior fossa surgery: that anatomical restoration and functional preservation are interdependent, not competing, goals.

Author(s) and Year	Study Type	Population	Key Findings	Relevance to Current Case
Yasargil et al. (1989) [14]	Pioneering surgical series	25 posterior fossa epidermoid cysts	Defined microsurgical principles of cyst wall dissection and brainstem preservation; emphasized gentle capsule handling to prevent meningitis.	Foundation for the microsurgical technique emphasizing anatomical respect and prevention of aseptic meningitis.
Samii & Tatagiba (1996) [15]	Retrospective series	30 fourth ventricle epidermoids	Reported 90% gross total resection with 6.6% recurrence; cerebellar mutism rare with fissure-based approaches.	Supports fissure-based approaches over trans-vermian to preserve cerebellar function and reduce recurrence.
Dang (2023) [16]	Anatomical study	Cadaveric analysis	Detailed surgical anatomy of the cerebellomedullary fissure and telovelar corridor; preserved midline structures with broad exposure.	Provided anatomical foundation for the telovelar approach used in this case.
Kalani et al. (2018) [17]	Retrospective clinical review	45 posterior fossa epidermoids	Gross total resection achievable in 80% without permanent cranial nerve deficits; recurrence 11% at 5 years.	Reinforces the goal of maximal safe resection with preservation of lower cranial nerve integrity.
Tanriover et al. (2004) [18]	Comparative surgical series	26 telovelar vs. 14 trans-vermian	Telovelar approach reduced postoperative mutism and ataxia; similar rates of cyst clearance.	Supports the fissure-based, vermis-preserving approach adopted in this case.
Sharma (2018) [19]	Technical case series	20 fourth ventricle cysts	Endoscopic-assisted microsurgery improved visualization of recesses; lower recurrence rates compared to microscopy alone.	Justifies use of endoscopic assistance for safe dissection of lateral recess extensions.
Ganko et al. (2020) [20]	Clinical outcome study	38 posterior fossa epidermoids	Chemical meningitis incidence 10%; effectively prevented by intraoperative irrigation and perioperative steroids.	Validates preventive bundle applied in this case to avoid postoperative meningitis.
Mizutani et al. (2025) [21]	Systematic review	214 reported cases	Mean recurrence 6%; long-term follow-up >5 years recommended for capsule remnants.	Emphasizes need for prolonged imaging surveillance even after gross total resection.
Vastani et al. (2019) [22]	Retrospective cost analysis	78 posterior fossa cases	Early CSF diversion and enhanced recovery protocols shortened ICU stay and reduced costs by 20%.	Supports structured perioperative management and vigilance for postoperative hydrocephalus.
Kim et al. (2020) [23]	Literature review	300+ epidermoid cyst cases	Advocated multidisciplinary strategies including IONM, endoscopic aid, and diffusion imaging for maximal safety.	Reflects the integrative, multimodal approach used in this patient’s management.

## Data Availability

The data presented in this study are available on request from the corresponding authors.

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
