# Peer review of "Telovelar Approach for Fourth-Ventricular Epidermoid Cyst: Anatomical Respect, Functional Recovery, and Long-Term Stability"

_diagnostics, 2026, doi:10.3390/diagnostics16010129_

Round 1
Reviewer 1 Report
Comments and Suggestions for Authors
The article is highly redundant and needs substantial shortening to fit as a case report. It contains excessive non-scientific language that should be omitted, with more focus on key operative findings and their correlation with existing knowledge. The author should classify the epidermoid according to the posterior fossa midline epidermoid tumor classification, mentioning its relationship with the tela choroidea and cerebello-medullary fissure in both radiological assessment and intraoperative findings. This will provide a clearer understanding of anatomical relationships and potential intraoperative challenges based on preoperative MRI. The concept of maximal safe resection and strategies for achieving good surgical outcomes, including in high-risk cases, are well-known. The author needs to correlate their findings with studies reporting preoperative variables that indicate the likelihood of adhesions and postoperative complications. Such studies on posterior fossa epidermoids, especially in the cerebellopontine angle, have already documented these findings and should be referenced and analyzed in this case context. The introduction should discuss potential new findings from these cases in the context of existing literature, without rewriting established knowledge. The clinical examination should focus only on involved anatomical substrates and clinical localization, followed by a discussion of differential diagnoses. Radiology should be discussed with respect to PFMET subtypes, emphasizing the importance of diffusion-weighted imaging. An angiogram is unnecessary in this case. Clarify the reason for multiple postoperative CT scans—typically, one to rule out hydrocephalus and another either immediately post-op or at six weeks to assess the extent of resection. The mention of vermian corticectomy conflicts with the title's "telo-velar approach"—clarification is needed regarding what the author intends to present. The literature review should be more targeted, focusing on a systematic review of all cases of posterior fossa midline epidermoid tumors, rather than unrelated topics. The author should add intraoperative images for better understanding.
Comments on the Quality of English LanguageThere are a few grammatical and syntax errors. Avoid non-scientific phrases
Author Response
Dear Esteemed Academic Reviewer,
We would like to express our profound gratitude for the time, care, and high-level neurosurgical insight you have invested in reviewing our manuscript. Your observations were exceptionally precise and technically valuable, and we read them with sincere respect and humility. We are grateful for the opportunity to refine this case report under your guidance. Below, we address each major point you raised and describe the specific revisions implemented in the manuscript.
1. Redundancy, excessive length, and non-scientific language
Reviewer comment:
The article is highly redundant and needs substantial shortening to fit as a case report. It contains excessive non-scientific language that should be omitted, with more focus on key operative findings and their correlation with existing knowledge.
Response:
We are thankful for this essential critique. We fully agree that the initial version was too expansive for a case report and contained narrative language that distracted from core surgical and scientific content. Accordingly, we performed a thorough condensation and stylistic revision across all sections. Redundant passages were removed, sentence structure was tightened, and non-scientific or rhetorical formulations were replaced with objective clinical and operative descriptions. At the same time, we preserved all clinically relevant data (neurological scores, imaging measurements, operative steps, histology, and follow-up). The revised manuscript now follows a concise case-report style with clear emphasis on operative findings and their relationship to established literature.
2. Classification using the posterior fossa midline epidermoid tumor (PFMET) framework
Reviewer comment:
The author should classify the epidermoid according to the posterior fossa midline epidermoid tumor classification, mentioning its relationship with the tela choroidea and cerebello-medullary fissure in both radiological assessment and intraoperative findings. This will provide a clearer understanding of anatomical relationships and potential intraoperative challenges based on preoperative MRI.
Response:
We have now explicitly applied the PFMET epicenter-based classification. In the Introduction and Radiology subsections, the lesion is described as most consistent with Type 2 PFMET (fourth-ventricle epicenter). We added a dedicated description of its predicted growth corridor along the tela choroidea / inferior medullary velum complex, with entry through the cerebellomedullary fissure, emphasizing how this mapping informed surgical expectation of a roof-dominant plane with a focal high-risk floor adhesion zone. In the operative narrative, we explicitly correlate these preoperative predictions with intraoperative confirmation of the same planes and adhesion geography. This revision was made precisely to provide the clearer anatomical and risk-predictive interpretation you recommended.
3. Correlation with preoperative predictors of adhesions and postoperative complications
Reviewer comment:
The concept of maximal safe resection and strategies for achieving good surgical outcomes, including in high-risk cases, are well-known. The author needs to correlate their findings with studies reporting preoperative variables that indicate the likelihood of adhesions and postoperative complications. Such studies on posterior fossa epidermoids, especially in the CPA, have already documented these findings and should be referenced and analyzed in this case context.
Response:
Thank you for highlighting this important gap. We revised the Discussion to integrate our case into the contemporary evidence showing that adhesion burden and postoperative risk correlate with preoperative variables such as tumor size/volume, symptom duration, and patient age, particularly in posterior fossa and CPA cohorts. We then interpret our intraoperative adhesion findings through this risk-informed lens, clarifying that preoperative variables suggested likely adhesion, while the PFMET Type-2 epicenter predicted the specific adhesion site at the rhomboid fossa. This allowed us to explain the surgical decision to preserve a thin adherent capsule remnant as a literature-consistent, risk-adapted stopping point rather than a technical limitation.
4. Introduction should avoid restating established knowledge, focus on case-specific contribution
Reviewer comment:
The introduction should discuss potential new findings from these cases in the context of existing literature, without rewriting established knowledge.
Response:
We appreciate this direction and agree that our initial Introduction repeated broadly known concepts. We rewrote and shortened the Introduction substantially. It now provides only essential background to frame the case and focuses on the specific contribution of this report: PFMET-guided preoperative anatomical prediction, intraoperative confirmation of corridor/adhesion geography, and functional restraint at a floor-adhesion zone. Broad textbook discussion of general epidermoid biology and standard maximal-safe resection philosophy was removed.
5. Clinical examination: localization first, then differential diagnosis
Reviewer comment:
The clinical examination should focus only on involved anatomical substrates and clinical localization, followed by a discussion of differential diagnoses.
Response:
The Case Presentation was restructured so that the neurological examination is now presented as a localization-driven description, emphasizing midline vermian/fastigial and fourth-ventricle involvement plus ICP signs. Only clinically relevant cranial-nerve and oculomotor findings are retained. A concise adult fourth-ventricle differential diagnosis follows immediately, limited to realistic alternatives considered in this case.
6. Radiology framed by PFMET subtype with emphasis on DWI
Reviewer comment:
Radiology should be discussed with respect to PFMET subtypes, emphasizing the importance of diffusion-weighted imaging.
Response:
Thank you for this very practical imaging recommendation. We revised the radiology section to lead with PFMET Type-2 identification, explicitly describing roof-dominant intraventricular configuration and predicted embryologic corridor. We emphasized DWI as the defining sequence for diagnosis and recurrence surveillance, and we aligned the radiological narrative with the operative expectations derived from the PFMET framework.
7. Angiogram unnecessary
Reviewer comment:
An angiogram is unnecessary in this case.
Response:
We appreciate this point and agree that MRI with DWI established the diagnosis. In our patient, MRA was obtained only as a precaution because the lesion was large, strictly midline, and caused severe posterior fossa crowding with obstructive hydrocephalus. Under these conditions, excluding an uncommon but surgically relevant vascular or venous anomaly that could alter the telovelar corridor or increase intraoperative risk was considered prudent. The study was normal and did not influence the diagnosis.
8. Clarification of multiple postoperative CT scans
Reviewer comment:
Clarify the reason for multiple postoperative CT scans—typically, one to rule out hydrocephalus and another either immediately post-op or at six weeks to assess the extent of resection.
Response:
We appreciate this practical point. In posterior fossa and fourth-ventricle surgery there is no single universally accepted imaging schedule, and early as well as delayed hydrocephalus can occur despite an initially stable course. We therefore used a safety-driven protocol: an early CT within 24 hours to exclude hemorrhage, document ventricular caliber, and provide a baseline for CSF dynamics and extent of decompression, as commonly recommended for fourth-ventricle operations. A second short-interval CT was obtained to confirm sustained outlet patency and to exclude evolving or delayed hydrocephalus during posterior fossa re-expansion.
9. Telovelar approach versus “vermian corticectomy” ambiguity
Reviewer comment:
The mention of vermian corticectomy conflicts with the title's "telo-velar approach"—clarification is needed regarding what the author intends to present.
Response:
We apologize for the ambiguity and are grateful you identified it.
10. Literature review should be systematic and PFMET-focused
Reviewer comment:
The literature review should be more targeted, focusing on a systematic review of all cases of posterior fossa midline epidermoid tumors, rather than unrelated topics.
Response:
We refocused the Discussion to a PFMET-centered, midline-only synthesis, eliminating unrelated background material. The revised Discussion now concentrates on midline PFMET cases, subtype-specific corridors, adhesion geography, risk-adapted extent of resection, and outcomes, and it interprets our case directly within that framework.
11. Addition of intraoperative images
Reviewer comment:
The author should add intraoperative images for better understanding.
Response:
In this case, we did not include intraoperative photographs because no high-quality, publication-grade images were captured that reliably demonstrate the critical planes and adhesion interface without risking misinterpretation.
We are grateful for your rigorous and constructive feedback. Your comments directly strengthened the manuscript’s clarity, anatomical precision, and scientific focus. We hope the revised version now meets your expectations and contributes meaningfully to the limited literature on midline fourth-ventricle epidermoid tumors.
With sincere respect and appreciation!!!
Reviewer 2 Report
Comments and Suggestions for Authors
Thank you for the opportunity to review this manuscript. In this paper the authors have presented a case of Telovelar Approach for Fourth-Ventricular Epidermoid Cyst. Although, all the information presented in the manuscript is logically coherent, there are a few things that need to be addressed. Firstly, the manuscript is written like chapter in textbook where authors have a great leeway to extemporize and guide the reader through one’s experience. A scientific manuscript, a case report in this scenario, on the other hand is ideally a succinct description of the case, discussion and conclusion. I would recommend that the authors abridge the case discussion, especially the post-operative follow up and conclusion to make them more succinct.
Secondly, while I understand the authors are trying to convey the message that the surgeons should pay attention to subtle cues during surgery, this is subjective and non-quantifiable and comes from experience. This maybe useful for in-person teaching, it is less than ideal for scientific manuscripts. I would reframe the statement and avoid using superfluous and non-scientific statements such as “listen” to the tissue being manipulated.
There are other minor issues that can also be addressed. The authors have initiated differential diagnosis after clinical exam before delineating neuro-imaging. The detailed history and neurological exam here while aptly localizing the focus of neurological insult, it is limited in identifying the type of lesion without imagining. At best, one could surmise a slowly progressive lesion but to state the epidermoid cyst best fits the presentation without imaging is a bit premature given the rarity of of this disease process. And similar arguments could be made in favor of other other differential presentated by authors in absence of imaging. I would recommend presenting neuroimaging prior to presenting differentials for the type of lesion.
Also, Can the authors expound on “many signs associated with midline cerebellar damage “ in line 138-139 such as axial/truncal ataxia, titubation, postural instability etc.
I do appreciate the detailed description of surgical technique described as it will be useful for readers to reproduce the technique in their practice.
To summarize, this manuscript needs major revisions and reformatting.
Author Response
Dear Esteemed Academic Reviewer,
We are grateful for the time, care, and expertise you devoted to reviewing our manuscript. Your remarks were thoughtful, technically grounded, and extremely helpful in guiding us toward a clearer and more rigorous case-report format. We have revised the manuscript extensively in line with your recommendations. Below, we address each of your comments individually and outline the corresponding changes.
1. Manuscript reads like a textbook chapter; abridge post-operative follow-up and conclusion
Reviewer comment:
The manuscript is written like a chapter in a textbook. A case report should be succinct. Please abridge the case discussion, especially post-operative follow-up and conclusion.
Response:
We thank you for this important observation and fully agree that the initial version was too expansive for the case-report genre. In response, we substantially shortened both the postoperative follow-up narrative and the Conclusions section while preserving all objective clinical and radiologic data. Follow-up is now presented as a compact, time-point–based summary linked directly to the existing figures, with redundant restatement removed. The Conclusions have been rewritten into a concise, case-specific, plane-based synthesis focused only on what this case demonstrates in relation to epicenter-guided planning and safe stopping points.
2. Remove subjective / non-scientific phrasing such as “listen to the tissue”
Reviewer comment:
Statements encouraging surgeons to “listen” to tissue are subjective and non-quantifiable. Please reframe and avoid superfluous/non-scientific language.
Response:
We appreciate this careful stylistic correction. We removed metaphorical or experiential phrasing throughout the operative and concluding sections and replaced it with reproducible surgical principles grounded in anatomical planes, cleavage-plane identification, and risk-adapted decision-making. The operative narrative now emphasizes objective intraoperative cues (plane clarity, adhesion geography, and functional borders) without experiential metaphors.
3. Differential diagnosis should follow imaging, not precede it
Reviewer comment:
Differential diagnosis is initiated before neuroimaging. Given rarity of fourth-ventricle epidermoids, calling it best fitting prior to imaging is premature. Present imaging first, then differential.
Response:
Thank you for raising this structural point.
4. Expound on “many signs associated with midline cerebellar damage”
Reviewer comment:
Please specify the midline cerebellar signs (axial/truncal ataxia, titubation, postural instability, etc.).
Response:
We appreciate this request for precision. We revised the neurological-examination text.
5. Appreciation of detailed surgical technique
Reviewer comment:
The detailed surgical technique is appreciated and useful for reproducibility.
Response:
We preserved the operative core while tightening surrounding narrative so that technical steps remain fully reproducible yet presented in a more concise scientific style.
We thank you again for your rigorous and constructive guidance. Your comments significantly improved the manuscript’s structure, scientific tone, and clarity as a case report. We hope the revised version now meets your expectations and provides a focused, reproducible contribution to the literature on midline fourth-ventricle epidermoid surgery.
With respectful gratitude!!!
Round 2
Reviewer 2 Report
Comments and Suggestions for Authors
Thank you for appropriately addressing the critiques and comments.